# Acute Vertigo, Dizziness and Imbalance in the Emergency Department—Beyond Stroke and Acute Unilateral Vestibulopathy—A Narrative Review

**DOI:** 10.3390/brainsci15090995

**Published:** 2025-09-15

**Authors:** Sun-Uk Lee, Jonathan A. Edlow, Alexander A. Tarnutzer

**Affiliations:** 1Neurotology and Neuro-Ophthalmology Laboratory, Korea University Medical Center, Seoul 02841, Republic of Korea; sulee716@gmail.com; 2Department of Neurology, Korea University Medical Center, Seoul 02841, Republic of Korea; 3Emergency Medicine, Beth Israel Deaconess Medical Center, Boston, MA 02215, USA; jedlow@bidmc.harvard.edu; 4Emergency Medicine, Harvard Medical School, Boston, MA 02115, USA; 5Neurology, Cantonal Hospital of Baden, 5404 Baden, Switzerland; 6Faculty of Medicine, University of Zurich, 8006 Zurich, Switzerland

**Keywords:** diagnostic approach, acute vestibular syndrome, episodic vestibular syndrome, differential diagnosis, thiamine deficiency, intoxication, electrolyte disturbances

## Abstract

New-onset vertigo, dizziness and gait imbalance are amongst the most common symptoms presenting to the emergency department, accounting for 2.1–4.4% of all patients. The broad spectrum of underlying causes in these patients cuts across many specialties, which often results in diagnostic challenges. For patients meeting the diagnostic criteria for acute vestibular syndrome (AVS, i.e., presenting with acute-onset prolonged vertigo/dizziness with accompanying gait imbalance, motion intolerance, nausea/vomiting, with or without nystagmus), the typical differential diagnosis is vertebrobasilar stroke and acute unilateral vestibulopathy. However, other disorders may also present with AVS. These include non-neurological causes such as drug side-effects or intoxication, electrolyte disturbances, cardiac disease, severe anemia, carbon monoxide poisoning, endocrine disorders and others. Other non-stroke neurological disorders may also present with AVS or episodic vertigo/dizziness, including demyelinating CNS diseases, posterior fossa mass lesions, acute thiamine deficiency and vestibular migraine. Furthermore, acute physiological abnormalities (e.g., hypotension, fever, severe anemia) may unmask previous vestibular impairments that had been well-compensated. Here, we review the diagnostic approach to patients with acute-onset dizziness in the emergency room and discuss the most important differential diagnoses beyond stroke and acute unilateral vestibulopathy.

## 1. Introduction

Acute vertigo, dizziness and gait imbalance are amongst the most common causes of emergency department (ED) visits, with 2.1–4.4% of all ED admissions being attributed to acute vertigo or dizziness [1,2,3]. This translates to approximately 4.4 million consultations per year in the US (and 50 to 100 million worldwide [4]), resulting in estimated annual health care costs of over USD 10 billion in the US alone [5]. The underlying causes of acute vertigo, dizziness or imbalance vary greatly and encompass many specialties, including internal medicine, psychiatry, endocrinology and gastroenterology. Thus, the diagnosis poses a significant challenge for the emergency physician. Based on a review of 9472 patients presenting to a US ED due to acute dizziness between 1993 and 2005 from the National Hospital Ambulatory Medical Care Survey (NHAMCS), half of the patients had an underlying systemic condition (not vestibular or cerebrovascular) [1]. Approximately 15% of these systemic conditions are considered dangerous and potentially life-threatening but treatable (Table 1). Furthermore, some medical conditions commonly present with episodic dizziness or vertigo, such as cardiac arrhythmia, orthostatic hypotension (OH) and transient hypoglycemia [6].

Understanding the common neurological and neurotological causes of acute dizziness/vertigo can be the framework for triaging the patients. Acute unilateral vestibulopathy (AUVP) can be a common cause. For those accounts of acute dizziness for the majority of the population, nearly 5–10% of patients with posterior circulation stroke can also present with acute dizziness [7]. Benign paroxysmal positional vertigo, Menière disease or vestibular migraine usually present with triggered or recurrent vestibular syndrome, but it can often confuse the diagnosis at the initial presentation. Catching the nystagmus and neurotological signs indicating peripheral or central vestibulopathy is essential for the diagnosis in these cases [8,9,10].

Thus, for patients presenting to the ED with acute dizziness, a rapid and thorough diagnostic approach is essential. While new-onset or recurrent vertigo, dizziness or gait imbalance is the leading symptom of the spectrum of disease discussed here, knowledge about accompanying signs and symptoms, conditions triggering or worsening complaints and pre-existing conditions, including current medication, is essential in the diagnostic workup. Notably, clinicians should prioritize the most prevalent and potentially dangerous conditions. Additionally, diagnostic algorithms that facilitate narrowing down the differential diagnosis have been reported. Among these, the TiTrATE approach has caught most attention, proposing to focus on a few elements of history taking (Timing and Triggers) and a Targeted Examination [4]. This approach is really no different than the approach to any symptom—defining its temporal characteristics and associated symptoms and then using the physical examination to narrow the differential diagnosis. Therefore, a general neurologic examination should be combined with the search for subtle ocular motor signs, an assessment of stance and gait, hearing and positional testing. While the presence of severe gait and truncal instability, central ocular motor signs (including gaze-evoked nystagmus and skew deviation) or new-onset unilateral hearing loss makes central (mostly ischemic) vestibular causes more likely; triggered nystagmus on positional testing favors benign paroxysmal positional testing. Likewise, absence of ocular motor, vestibular abnormalities and focal neurologic signs makes non-neurological causes more likely (as discussed in the following sections).

**Table 1 brainsci-15-00995-t001:** Differential Diagnosis of AVS based on expert opinion (modified after [7,11]).

Benign * or Less Urgent Causes		Dangerous * and More Urgent Causes	
Neurological Causes	Non-Neurological Causes	Neurological Causes	Non-Neurological Causes
*Common Causes (>1% of AVS)* acute unilateral vestibulopathy demyelinating disorders (e.g., multiple sclerosis, NMOSD, MOGAD) viral labyrinthitis *Uncommon (<1%) or Unknown Frequency* acute disseminated encephalomyelitis acute traumatic vestibulopathy/labyrinthine concussion antiepileptic drugs (e.g., phenytoin, carbamazepine, oxcarbazepine, but also lacosamide, gabapentin, pregabalin) anti-arrhythmic drugs (e.g., amiodarone) benzodiazepines cerebello-pontine angle neoplasm (e.g., vestibular schwannoma, metastases) episodic ataxia syndrome attack herpes zoster oticus (Ramsay Hunt) medication ototoxicity (e.g., aminoglycosides) prolonged Menière syndrome attack prolonged vestibular migraine attack *Presumed Possible Causes ^‡^* degenerative cerebellar ataxia intoxication (e.g., alcohol, illicit substances) unusual presentation of infection (otosyphilis, lyme borreliosis) *Uncertain Causes* musculoskeletal causes (e.g., cervicogenic dizziness, temporomandibular disorders)	*Common Causes* acute episode of major depression essential hypertension OH/POTS panic attack *Uncommon or Unknown Frequency* adrenal insufficiency, type 1 diabetes mellitus, diabetes insipidus alcohol withdrawal chemotherapeutic agents (cytarabine, epothilone D drug withdrawal (e.g., opioids, antidepressants) industrial chemicals (e.g., toluene, pesticides)	*Common Causes (>1% of AVS)* brainstem or cerebellar stroke (ischemic/hemorrhagic) *Uncommon (<1%) or Unknown Frequency* atypical anti-GQ1b antibody syndrome (AVS variant) autoimmune inner ear disease/autoimmune vestibulopathy (e.g., Cogan syndrome) bacterial labyrinthitis/mastoiditis brainstem encephalitis or cerebellitis (e.g., autoimmune/paraneoplastic, listeria, herpes simplex/zoster, paraneoplastic, Creutzfeldt-Jakob disease) labyrinthine stroke ^†^ neuro-oncologic emergencies (e.g., increased ICP, brain herniation, tumor bleeding) Wernicke syndrome (vitamin B1 deficiency) *Presumed Possible Causes ^‡^* anterior-circulation stroke CNS medication toxicity (e.g., lithium) hypertensive encephalopathy subarachnoid hemorrhage/aneurysm	*Common Causes* cardiac arrhythmia *Presumed Possible Causes ^‡^* carbon monoxide poisoning ciguatera poisoning decompression sickness electrolyte disturbances (e.g., hyponatremia, hypomagnesemia) endocrine disorders (e.g., hypoglycemia, hyperglycemia, hypothyroidism) leukemia, polycythemia vera, essential thrombocytosis or disseminated intravascular coagulation mountain sickness severe anemia or hypoxia

AVS = acute vestibular syndrome, CNS = central nervous system, ICP = intracranial pressure, MOGAD = myelin oligodendrocyte glycoprotein antibody-associated disease, NMOSD = neuromyelitis optica spectrum disorder, OH = orthostatic hypotension, POTS = postural tachycardia syndrome. * Any condition causing dizziness/vertigo can be considered a ‘dangerous’ medical problem if symptoms occur in dangerous circumstances (e.g., highway driving or free-rock climbing). Furthermore, the high vagal tone that accompanies some vestibular disorders can provoke bradyarrhythmias in susceptible individuals, including during the examination [12]. Nevertheless, although they may be quite disabling to patients during the acute illness phase, diseases classified here as ‘Benign or Less Urgent Causes’ rarely produce severe, irreversible morbidity or mortality (unlike their ‘Dangerous’ counterparts). ^†^ The frequency of labyrinthine infarction is difficult to estimate given that the current reference standard test for confirming the diagnosis (i.e., autopsy with temporal bone histology) is rarely performed. Recent studies, however, have suggested that patients with sudden deafness, with or without vertigo, are at increased risk of stroke, implying a vascular mechanism [13,14]. ^‡^ “Presumed possible causes” are conditions known to cause acute dizziness, but it remains unknown whether they can present with a clinically complete or clinically predominant AVS picture.

Vestibular syndromes should be classified according to the diagnostic approach proposed by the Bárány Society [15], which distinguishes three principal syndromes based on their (re)occurrence (acute, episodic and chronic) and whether episodes are triggered or occur spontaneously. Acute vestibular syndrome (AVS) is defined as a rapid onset of dizziness or vertigo, nausea or vomiting, head motion intolerance or gait instability, with or without nystagmus, lasting for at least 24 h [16]. Of course, patients who present before the 24 h threshold with ongoing dizziness should be managed as if they have AVS. The term “acute imbalance syndrome” (AIS) is often used to describe patients presenting with gait imbalance in the absence of prominent vertigo or dizziness or nystagmus.

Episodic vestibular syndrome (EVS) describes intermittent dizziness lasting for seconds, minutes or hours [4]. Such patients mostly experience multiple, discrete episodes spaced out over time. EVS is often further classified by the presence of a reproducible trigger (the triggered EVS [t-EVS], e.g., such as benign paroxysmal positional vertigo—BPPV) versus those with the absence of a trigger (the spontaneous EVS [s-EVS], e.g., due to vestibular migraine). A discussion of chronic vestibular syndrome (CVS) that includes persistent vertigo or dizziness over extended periods is beyond the scope of this review.

## 2. Methods

We searched MEDLINE through PUBMED for English-language articles based on the following strategy and looking for specific components in all articles: (1) acute/new-onset or recurrent dizziness/vertigo caused by (2) etiology other than central or peripheral vestibular origin. There was no formal rating or study quality assessment of identified citations, and we did not apply pre-specified inclusion or exclusion criteria. Only original data on human subjects were included for this review. We did not limit our search to a specific publication period (e.g., last 10 years). A manual search of the references of articles discussed here was also performed. Research abstracts from meeting proceedings or unpublished studies were omitted.

## 3. Non-Neurological Causes of AVS

Various benign and dangerous non-neurological conditions may present with acute-onset persistent or acute and episodic dizziness, vertigo or gait imbalance. In this review, we discuss the most frequent and the most dangerous conditions observed, focusing on the key clinical features and the differential diagnosis.

### 3.1. Pharmacologic Intoxication, Drug Withdrawal and Psychiatric Disorders

Pharmacologic intoxications may be caused by therapeutic medications, alcohol or illicit drugs. Acute vertigo, dizziness and gait imbalance are common medication side-effects that may accompany other symptoms, including altered consciousness, motor weakness or sensory changes. Unilateral vestibular involvement causing AVS is rare. For example, the intake of phosphodiesterase inhibitors to treat erectile dysfunction may result in possible unilateral inner-ear ischemia [16].

However, therapeutic or supra-therapeutic levels of various medications that affect the CNS can cause acute reversible vertigo, dizziness and gait imbalance [11]. These include antiepileptic drugs often used to treat epilepsy or neuropathy (especially phenytoin, carbamazepine [17] and oxcarbazepine but also lacosamide, gabapentin and pregabalin) [18], benzodiazepines and lithium. Additionally, chemotherapeutic agents, such as cytarabine or epothilone D [18], and anti-arrhythmic drugs, such as amiodarone [19], may result in acute-onset vertigo, dizziness or gait imbalance. In a meta-analysis of patients presenting with AVS, one case (out of 422) was caused by carbamazepine intoxication [20], and in a case series, one of 16 cases with (sub)acute bilateral vestibulopathy (presenting as AVS) was caused by phenytoin intoxication [21]. Although this indicates that drug intoxication is a relatively infrequent cause of central AVS, prior studies may have focused on (ischemic) central AVS cases, potentially under-estimating its occurrence.

Another important differential diagnosis in patients with acute dizziness and ataxia is alcohol poisoning, which typically presents with varying degrees of altered mental status, behavioral abnormalities, slurred speech, gait ataxia and vertigo or dizziness due to both peripheral vestibular and central (cerebellar) mechanisms. Various patterns of nystagmus, including positional nystagmus [22], can be observed; horizontal gaze-evoked nystagmus due to impaired eccentric gaze holding is the most common nystagmus observed [18,23].

Symptoms typically resolve within hours to days after the causative substance is discontinued. However, for certain drugs, very high doses and/or prolonged exposure may result in permanent damage to the vestibular organs (as seen with gentamicin and heroin) or the cerebellum (as seen with phenytoin, phenobarbital, carbamazepine and alcohol) [18,24]. Bilateral vestibulopathy caused by aminoglycoside-related vestibular hair cell damage [25] may manifest rapidly with acute vertigo accompanied by hearing loss [26] or emerge slowly in these situations, with gait imbalance being the presenting symptom [18,24,27]. Patients with (sub)acute-onset gait ataxia due to nitrous oxide abuse and resulting acute myelopathy and sensorimotor neuropathy typically show minimal short-term improvement but remarkable long-term recovery [28].

Additionally, abrupt cessation of certain drugs may result in acute vertigo or dizziness, typically delayed by 3–5 days and potentially lasting weeks [29,30,31]. The less abrupt onset and accompanying autonomic signs and symptoms (as in alcohol, antidepressants or opiate withdrawal) may help differentiate drug-withdrawal-related vertigo from vertebrobasilar stroke or AUVP [11].

Vertigo and dizziness are also associated with psychiatric disorders. Most frequently, these are chronic conditions such as persistent postural perceptual dizziness in patients with general anxiety or depression. However, acute or episodic vertigo and dizziness may be observed with (recurring) panic attacks or depression [32], which accounts for 1.1% of all dizziness cases in a large ER sample based on the NHAMCS data (1993–2005) [1]. In another study analyzing the NHAMCS data over 10 years (1995–2004), approximately 7–8% of all dizzy patients presenting to the ER suffered from various psychiatric disorders [33]. The lack of accompanying nystagmus, absence of motion intolerance, a history of a psychiatric disorder and generally more chronic nature of the symptoms help to distinguish neurologic disorders from psychiatric causes in these patients.

### 3.2. Environmental Toxins

Numerous industrial chemicals, such as toluene or pesticides (including organophosphates), can cause vertigo or dizziness, which may have an acute onset [34]. More specific symptoms such as salivation, lacrimation or diarrhea due to muscarinic effects of organophosphate intoxication help to differentiate from vestibular causes of acute vertigo or dizziness in these cases. Headache and dizziness are the most commonly reported complaints of carbon monoxide exposure, which may account for more than half of all fatal poisonings worldwide [35]. Noteworthy, more specific symptoms may not manifest early and emerge only with increasing carbon monoxide exposure (including altered mental status, seizures, coma and stroke-like symptoms) [35]. A single case of carbon monoxide poisoning with nystagmus masquerading as a vertebrobasilar stroke or AUVP has been reported [36]. Such exposures are often suspected through patient history.

### 3.3. Endocrine Disorders

Endocrine disorders and resulting impairment of blood glucose or thyroid hormone levels may cause various vestibular syndromes. These include both AVS and t-EVS, depending on the underlying mechanism. For example, in cases with adrenal insufficiency, type 1 diabetes or diabetes insipidus, vertigo or dizziness can result from OH and is thus episodic and triggered by body position. In contrast, patients with severe hypothyroidism [37] or acute hyperthyroidism may present with acute and persistent dizziness [38], nausea and vomiting [39]. It remains unknown whether the vestibular system is directly engaged in hypothyroidism. A recent publication on this topic focused on treated hypothyroidism and reported no ocular motor abnormalities besides positional nystagmus due to BPPV [40]. Importantly, persistent nausea or vomiting of any cause may lead to AVS secondary to acute thiamine (vitamin B1) deficiency as well (see below). Thus, vitamin B1 deficiency should be considered in the differential diagnosis in these cases [41,42].

Hypoglycemia is a well-known cause of dizziness and vertigo [43,44], accounting for 1.4% of all dizzy patients presenting to the ED in the large NHAMCS database [1]. In cases of transient hypoglycemia (due to overdose of insulin or oral hypoglycemic agents), dizziness is often accompanied by light-headedness, gait imbalance, sweating, tremulousness, generalized weakness and mental confusion [45]. In contrast, acute hyperglycemia seems to be a rare cause of acute persistent vertigo or dizziness (resembling a vertebrobasilar stroke or AUVP) [11], although patients may have OH due to an osmotic diuresis.

In patients with acute dizziness, it is important to consider the medical history and specifically ask questions concerning existing endocrinologic disorders, such as diabetes, hypoglycemic medications or thyroid disease. In patients with thyroid hormone replacement therapy, thyroid hormone and thyroid stimulating hormone levels should be checked to avoid missing cases with insufficient supplementation [37] or hyperthyroidism [46].

### 3.4. Electrolyte Disturbances

Disturbances in fluid levels or blood electrolytes were identified in 5.6% of all dizzy patients presenting to the ED in the large NHAMCS database [1]. Likewise, in another study including patients presenting to the ED with sudden-onset dizziness, 2% of cases had symptomatic hypocalcemia [47]. Volume depletion due to increased fluid loss (due to prolonged diarrhea, repetitive vomiting, increased sweating or renal dysfunction) or reduced fluid intake usually results in episodic dizziness rather than in acute prolonged vertigo or dizziness. This is due to the resulting OH on standing up (see the separate section on OH for details).

Hyponatremia is the most common electrolyte disorder associated with persistent dizziness. In symptomatic hyponatremia after pituitary surgery, dizziness (38%) and nausea or vomiting (29%) were the most frequently reported symptoms [48]. Thus, symptoms may be confused with vertebrobasilar insufficiency, whereas a distinction from AUVP is readily made owing to the absence of nystagmus. In hypernatremia, acute dizziness or vertigo may occur as a result of dehydration and OH rather than due to CNS-related symptoms of hyperosmolarity [49]. Similarly, hypotension due to bradycardia is the most plausible mechanism to explain vertigo or dizziness in patients with hypokalemia, such as those with Gitelman syndrome [50]. Likewise, hyperkalemia was associated with dizziness, presyncope or syncope in about 1/3 of patients (58/169, 34.3%) in a single prospective study [51]. A combination of hyperkalemia and medications that block the atrioventricular node may lead to synergistic bradycardia and may result in a variety of symptoms, ranging from asymptomatic bradycardia or dizziness to the full picture of BRASH syndrome (bradycardia, renal failure, atrioventricular blockade, shock and hyperkalemia) [52].

Other electrolyte changes including hypo- or hypercalcemia [53], hypo- or hyperphosphatemia and hypomagnesemia have also been linked to vertigo or dizziness, albeit the pathomechanisms remain unclear. The correlation is often vague as other dyselectrolytemias are frequently observed in these conditions. Moreover, these patients often have comorbidities, such as renal failure, thyroid dysfunction, fluid level changes or cardiac arrhythmia (as seen in hypocalcemia [54] or hypercalcemia [55]), that altogether prevent the linking of these symptoms to a certain pathomechanism.

Among the different electrolyte disturbances, abnormal eye movements were most consistently identified in patients with hypomagnesemia. A systematic review reported downbeat nystagmus (27%), horizontal nystagmus not further specified (18%) and slowing of horizontal saccades, which may be accompanied by cerebellar ataxia [56]. Various conditions can cause hypomagnesemia, including deficient magnesium intake, excessive loss through gastrointestinal or renal excretion, drugs (most frequently proton pump inhibitors) and endocrinologic disorders. These conditions may result in vertigo, dizziness and oscillopsia and may be accompanied by other electrolyte disturbances, such as hypokalemia [57]. Likewise, in a case report, tacrolimus-related hypomagnesemia caused downbeat nystagmus explained by transient cerebellar hypofunction [58].

In patients in whom the diagnosis is not clearly established by physical examination (e.g., a patient with AUVP), basic laboratory analysis of blood and an electrocardiogram (ECG) will identify this group of patients.

### 3.5. Orthostatic Hypotension

OH was identified in 0.6% of all dizzy patients presenting to the ED in the large NHAMCS database [1]. As discussed in the previous paragraph, this condition may result from various causes that lead to volume depletion and generalized cerebral ischemia, typically triggering position-dependent episodic vertigo or dizziness (i.e., emerging immediately after standing up). Orthostatic hypotension may be accompanied by rotatory vertigo and nystagmus in up to 30% of cases, as reported in a single prospective study evaluating two conditions known to trigger symptoms. Specifically, 10/33 patients with OH developed nystagmus during either the Schellong test or the squatting–standing test, with nystagmus beating direction being either purely downbeat or combined downbeat–horizontal with or without torsion [59]. Nystagmus duration was less than 60 s in 7/10 cases.

Vestibular autonomic failure following acute unilateral vestibulopathy was described in a small case series. Specifically, two patients reported postural light-headedness, and postural tachycardia was noted in both in the first month after AUVP [60]. This was explained by reduced otolith-organ-driven muscle sympathetic nerve function, which, in turn, caused venous pooling.

### 3.6. Nutritional Disturbances

Both thiamine (vitamin B1) deficiency [18] and vitamin B12 deficiency [61,62] may result in acute-onset vertigo, dizziness or gait ataxia. Among patients presenting to the ED with an acute or subacute onset of symptoms, thiamine (vitamin B1) deficiency is the most important differential diagnosis. Early signs of acute thiamine deficiency (also referred to as Wernicke encephalopathy [WE]) include gait imbalance, ataxia and subtle ocular motor signs [63], such as spontaneous vertical (initially upbeat and then later, downbeat) nystagmus, horizontal gaze-evoked nystagmus and bilaterally impaired horizontal vestibulo-ocular reflex [64,65,66]. Classic symptoms of WE, such as altered mental status and ophthalmoplegia, may become apparent later, and the clinical triad is often incomplete [67,68]. Thus, identifying eye movement abnormalities is paramount for early diagnosis and treatment of WE. Although classically associated with chronic alcohol use, many other conditions may lead to insufficient thiamine intake, including fasting, persistent nausea or vomiting due to any reason, cancer, hyperemesis gravidarum [69], bariatric surgery, starvation and others. With thiamine storage being sufficient only for about 18 days without new intake, symptoms may evolve rapidly [70]. Although pre-treatment thiamine levels should be sent, immediate, empiric high-dose (200–500 mg three times per day) intravenous supplementation of thiamine is essential if a nutritional deficit is suspected [67,69]. If treatment is initiated early, eye movement abnormalities will often recover rapidly, and neurocognitive deficits can be prevented [41,63]. Thus, in any patient presenting with (sub)acute-onset vertigo, dizziness or gait ataxia with the above risk factors, vitamin deficiencies should be promptly considered as a possible culprit, and high-dose supplementation of thiamine should be initiated immediately.

Vitamin B12 deficiency may lead to vertigo/dizziness or gait imbalance by several mechanisms, including sensory ataxia due to peripheral polyneuropathy or myelopathy (being rarely acute onset) and orthostatic hypotension due to anemia. Treatment focuses on vitamin B12 supplementation in these cases.

### 3.7. Cardiac Disorders

The prevalence of dizziness in patients with cardiovascular diseases is approximately 10%, and patients may report either non-spinning dizziness or vertigo [71]. In the large NHAMC database, 21% of the 9472 patients presenting to the ED with dizziness had a cardiovascular condition causing the dizziness [1]. The most plausible underlying pathomechanism leading to vertigo or dizziness in cardiovascular diseases and cardiac arrhythmias is a drop in blood pressure due to decreased cardiac output (see illustrative case in Figure 1).

In particular, patients with postural orthostatic tachycardia syndrome typically present with postural vertigo or dizziness, resolving quickly and entirely by recumbence [72]. However, patients with episodic arrhythmias that happen to occur while lying down and that are associated with episodic hypotension can have episodic dizziness while lying down. That said, it is highly unusual for cardiac disease to result in stable, sustained hypotension (without losing consciousness) in both lying and standing positions that manifests over days (as in AVS due to vertebrobasilar stroke or AUVP). Furthermore, motion intolerance and nystagmus caused by underlying cardiac disease are unlikely. However, in patients with recurrent spontaneous dizziness or vertigo, positionally dependent or exertion-related serious cardiac disorders should also be investigated, even when obvious cardiovascular symptoms such as chest pain or dyspnea are missing [71].

Noteworthy, serious and potentially life-threatening cardiovascular disease (such as aortic dissection or myocardial infarction), anaphylaxis, pulmonary embolism or severe internal bleeding may present with stable hypotension and persistent vertigo or dizziness. In these patients, pathologic nystagmus and/or worsening of symptoms by head motion are typically absent. This is in contrast to patients suffering from vertigo/dizziness due to vertebrobasilar stroke or AUVP, often demonstrating both findings prominently. Likewise, bradycardia either due to medication side-effects (as seen in betablockers or calcium antagonists) or due to transient complete heart block (or other cardiac causes of bradycardia enough to drop blood pressure) may result in either postural or persistent vertigo or dizziness.

Acute vertigo or dizziness attributed to unspecified essential hypertension as a presenting cause to the ED was found in approximately 7–8% of patients in another study analyzing the NHAMCS data over a 10-year period (1995–2004) [33] and in 5.6% of patients in a prospective observational study conducted in a single tertiary center [73]. However, in clinical practice, elevated BP is often the consequence of AVS, and it remains unclear whether a clear cut-off point exists that can directly precipitate AVS. Collectively, either too low or too high blood pressure may trigger acute vertigo or dizziness.

Standard ED practice will identify abnormal blood pressure, pulse rate and hypoxemia. In cases with no clear diagnosis on physical exam, laboratory analysis of blood will identify anemia and other hematologic disorders [11].

### 3.8. Rheologic Disorders and Respiratory Disorders

Small vessel disease may result in neurologic symptoms due to various causes, including strokes due to inflammatory vascular disease (vasculitis), rheologic abnormalities, fat emboli or gas bubbles. Hyperviscosity or microangiopathy, as observed in hematologic disorders such as leukemia, polycythemia vera, essential thrombocytosis, or disseminated intravascular coagulation, may lead to vertigo or dizziness due to occlusion and infarction of either peripheral [74] or central vestibular structures [75]. In labyrinthine infarction, MRI-DWI is usually normal, and distinction from labyrinthitis is challenging [76]. However, if serologic tests are taken routinely in the ED, chances of missed diagnoses of such hematologic disorders are low. Both fat embolism and decompression sickness may result in sudden vertigo or dizziness due to the involvement of peripheral or central vestibular structures with accompanying nystagmus, closely mimicking vertebrobasilar stroke or AUVP [77,78,79]. The context (i.e., bone fracture(s) or surgery, presence of hematologic abnormalities or after returning from a dive) often provides important diagnostic clues.

According to the NHAMCS data, an underlying respiratory disorder was identified in 11.5% of all dizzy patients presenting to the ED [1]. In contrast, a smaller retrospective study involving 2126 cases reported a respiratory disorder identified in only 1.1% of patients [80]. Importantly, 55% of the patients in the NHAMCS data received symptom-related diagnoses only, including altitude illness (also called mountain sickness) [81], carbon monoxide poisoning (see section “environmental toxins”) and pulmonary embolism [1]. In contrast, no further information was provided in the other study [80]. A history of recent high-altitude exposure and respiratory complaints such as shortness of breath, reduced blood oxygen saturation and altered mental status helps to narrow the differential diagnosis in these cases. Hyperventilation of any cause can also lead to episodes of dizziness.

## 4. Non-Stroke Neurologic Conditions

A broad range of disorders affecting the central nervous system may result in vertigo, dizziness and gait imbalance besides ischemic or hemorrhagic strokes, as shown in several large samples of acutely dizzy patients [1,33,80]. Symptom onset may also be acute, mimicking cerebrovascular disease. However, patients can also present with episodic vertigo or dizziness. The underlying pathomechanisms include demyelinating disease; posterior fossa tumors; encephalitis; vestibular migraine and increased intracranial pressure. In contrast to non-neurological conditions linked to vestibular symptoms, these conditions frequently present with subtle ocular motor abnormalities, including spontaneous, positional and gaze-evoked nystagmus, making their distinction from cerebrovascular causes more challenging. In addition, MRI-DWI may be false negative in about 20% of vertebrobasilar strokes in the first 24–48 h [20]. Thus, a negative early MRI-DWI does not exclude the possibility of vertebrobasilar stroke if the clinical presentation indicates otherwise. In such cases, it is recommended to order a follow-up MRI-DWI at least 48–72 h later.

### 4.1. Demyelinating Disease Including Multiple Sclerosis

Multiple sclerosis (MS), which is the most frequently diagnosed type of demyelinating CNS disease, accounts for approximately 5% of all central AVS cases [20]. However, other forms of demyelinating CNS disease may result in vertigo, dizziness, ocular motor abnormalities or gait imbalance as well [82]. These diseases include acute disseminated encephalomyelitis, neuromyelitis optica spectrum disease [83] and myelin oligodendrocyte glycoprotein antibody disease [84].

Dizziness is a frequent complaint of patients with MS, being reported by 35 to 54% overall [85,86]. Brainstem lesions are found in 68 to 72% of patients with MS, and up to 70% of patients present with cerebellar signs [87,88]. Depending on the lesion location and the temporal evolution of lesions, vestibular symptoms including vertigo, dizziness and gait imbalance may present acutely and persist (AVS pattern) or may be episodic and position-dependent (due to periventricular lesions resulting in central positional vertigo) or gaze-dependent (due to internuclear ophthalmoplegia or nuclear lesions of the 3rd, 4th or 6th nerve and resulting ocular vertigo). An illustrative case of a patient with sudden-onset gait ataxia and vertigo due to an MS plaque located in the left dorsolateral medulla is shown in Figure 2.

Most commonly, demyelinating lesions affecting the root entry zone of the 8th nerve and the medial vestibular nucleus are found when patients are presenting with AVS [90]. Lesions within the medulla oblongata, cerebellar peduncles, posterior pontine tegmentum and midbrain can also be identified, presenting with trivial central ocular motor signs (central HINTS [20]) [91].

Importantly, acute demyelinating lesions affecting the cerebellar peduncle may result in intense, persistent, positional vertigo and downbeat nystagmus [92] or purely torsional nystagmus [93]. These patients may be misdiagnosed with BPPV; however, the nystagmus beating direction is not in the plane of the stimulated canal, and they do not respond to suitable liberation maneuvers. These features do not fit into a diagnosis of BPPV but favor a central cause, as emphasized in a recent review [94].

New-onset vertigo or dizziness may also arise from various other nystagmus patterns seen in demyelinating CNS disorders, including acquired pendular nystagmus (typically beating in the horizontal plane and caused by focal brainstem lesions) and dissociated nystagmus in internuclear ophthalmoplegia [95].

Of course, patients with a history of demyelinating CNS disorders presenting with vertigo, dizziness or gait ataxia can also have other etiologies. In one case series of patients with MS, BPPV was more frequently identified as the underlying cause than new MS plaques (52% vs. 32%) [96]. Other differential diagnoses include Menière‘s disease, vestibular migraine and AUVP [97].

### 4.2. Posterior Fossa Tumors and (Other) Causes of Increased Intracranial Pressure

Neoplasms located in the posterior fossa may result in (sub)acute-onset vertigo, dizziness or gait ataxia and can directly affect structures that play an important role in the central processing of vestibular information, such as the vestibular nuclei in brainstem glioma [98], cerebellar peduncles or cerebellar vermis. The primary vestibular afferents can be affected by vestibular schwannomas or meningiomas in the cerebellopontine angle. While symptoms often develop slowly, they can also manifest or worsen abruptly [99] due to rapid growth of the mass lesion, tumor bleeding [100] or swelling following radiation therapy [101] as described for vestibular schwannoma. Edema surrounding a mass lesion or tumor growth exceeding the compensatory mechanisms of the brain can also result in sudden worsening of symptoms due to rapidly increasing intracranial pressure.

Brainstem and cerebellar metastases may be the first symptom of a tumor outside the neuraxis. Thus, the lack of a history of a neoplastic disorder does not exclude the possibility of an intracranial mass lesion. In one series of 59 patients, the diagnosis of tumor was made based on symptomatic cerebellar metastasis in 25% (15/59 cases) [102]. This trend is also demonstrated in the illustrative case (see Figure 3).

Headaches were reported in up to 50% of ischemic strokes in a prospective study [104], which was a key finding in patients with vertebral artery dissection [105]. However, headache is also a characteristic finding in increased intracranial pressure due to a mass lesion (e.g., tumor or hemorrhage) and in patients with vestibular migraine [106]. Thus, a combination of acute-onset vertigo, dizziness and headache indicates a central origin of vestibular symptoms in the patient presenting to the ED but does not allow a distinction between vascular and non-vascular causes or between serious causes and vestibular migraine.

Increased intracranial pressure due to cerebral vein thrombosis may present with acute-onset and prolonged vertigo, vomiting and occipital headache, mimicking ischemic stroke or AUVP [107]. Patients initially presented with headache (92%), nausea/vomiting (75%) and dizziness (71%) in idiopathic intracranial hypertension, whereas nystagmus was described in only 10% of patients [108]. The opposite condition, i.e., intracranial hypotension (e.g., after lumbar puncture or cervical chiropractic manipulation or in case of spontaneous intracranial hypotension), may present as a new-onset orthostatic headache associated with dizziness, vomiting and double vision [109,110].

### 4.3. Vestibular Migraine

Episodic dizziness or vertigo lasting minutes to days is a hallmark sign of vestibular migraine [106]. Importantly, the first episode of acute-onset and prolonged vertigo or dizziness may be misdiagnosed as vertebrobasilar stroke and has even been treated with intravenous thrombolysis [111]. Patients with vestibular migraine attacks typically present with central ocular motor signs, including gaze-evoked nystagmus, spontaneous nystagmus and a bilaterally normal head-impulse test, rendering the presumed diagnosis of vertebrobasilar stroke. Vice versa, vertebrobasilar stroke can be misdiagnosed as an episode of vestibular migraine [112].

In a systematic review of the underlying causes of AVS, 7% (29/422) of patients diagnosed with central AVS had vestibular migraine [20]. Rarely, a peripheral vestibular pattern with unilaterally decreased gain on head-impulse testing may be seen in patients with vestibular migraine, mimicking AUVP [113]. Furthermore, patients with VM can present with recurrent positional vertigo/dizziness accompanying positional nystagmus [114]. Thus, with no evidence of structural abnormalities on follow-up brain MRI-DWI, the diagnosis of vestibular migraine heavily relies on structured history taking. One reason that diagnosing vestibular migraine is important is that missed or delayed diagnoses are associated with long periods of time prior to correct diagnosis. These delays lead to overtesting and patient morbidity [115]. Various elements from the history and epidemiological context can help distinguish between vestibular migraine and posterior circulation TIA [116,117].

## 5. Limitations

Being a narrative review, this paper has several limitations that need to be considered. First, we did not perform a systematic review of the literature; therefore, there is a certain risk for selection bias. However, this approach, at the same time, allowed us to provide a broad overview of the topic, emphasizing the spectrum of possible clinical presentations and underlying causes. Second, we did not perform a formal assessment of the quality of studies cited in this review (such as QUADAS2 or similar); thus, some findings discussed here may have a low level of evidence, especially when originating from small case series or single case reports. Third, for certain underlying disorders or clinical presentations, the literature available is relatively old, with more recent publications lacking. Thus, diagnostic approaches described in these publications may have changed meanwhile, and new diagnostic tests may have significantly altered clinical practice.

## 6. Conclusions

A broad range of non-neurologic and non-stroke neurological conditions result in acute or episodic vertigo, dizziness or gait imbalance. To narrow down the differential diagnosis in the ED setting, structured history taking (including medication, comorbidities, timing and triggers of the dizziness, context and accompanying symptoms) is the key and should be followed by a targeted examination, including vital signs, an assessment of ocular motor signs, gait, positional testing and a general medicine examination. In the absence of a clear diagnosis, the laboratory workup should include complete blood counts, electrolytes and glucose levels. ECG and Schellong testing for OH should be considered as routine screening in these patients. Finding dangerous disorders is a priority, with rapidly worsening symptoms, deteriorating vital signs, impaired consciousness and emerging focal neurologic signs reflecting red flags. In central AVS, stroke mimics such as vestibular migraine attacks, acute intoxication with CNS-active drugs or acute thiamine deficiency may be difficult to readily identify at the bedside as ocular motor signs can be subtle. For non-neurologic conditions presenting as AVS, the absence of nystagmus and lack of motion intolerance and accompanying systemic features of the underlying disorder usually allow for a more reliable distinction. Peripheral and/or central vestibular disorders that usually present in a chronic manner can also aggravate rapidly and present as an AVS due to acute stressors such as hypoxia or fever. Evaluating the acutely dizzy patient in the ED setting remains challenging. The difficulty may stem from the broad range of underlying disorders, the complexity of diagnostic algorithms available and the technical demands of bedside tests and interpretation. Therefore, dedicated training in the management of the acutely dizzy patient for the ED physicians should be prioritized. Until such time that emergency clinicians receive training in the physical examination techniques used to diagnose patients with acute dizziness, vertigo and imbalance, wider use of telemedicine services, ideally with video-oculography, is recommended, although this too is difficult to implement at scale.

## Figures and Tables

**Figure 1 brainsci-15-00995-f001:**
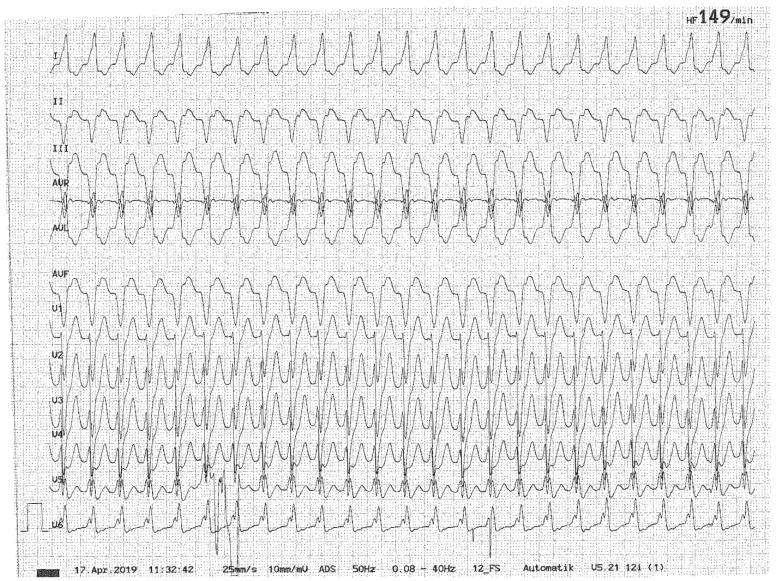
Illustrative case of a 52-year-old male patient presenting with new-onset, episodic spontaneous dizziness to the ED. On history taking, he also reported palpitations and an episode of near-fainting. Ventricular tachycardia was intermittently seen on ECG, which spontaneously converted into a sinus rhythm before initiating the treatment. With a history of cardiac infarction two weeks earlier, this cardiac arrhythmia was most likely related to a cardiac tissue scar.

**Figure 2 brainsci-15-00995-f002:**
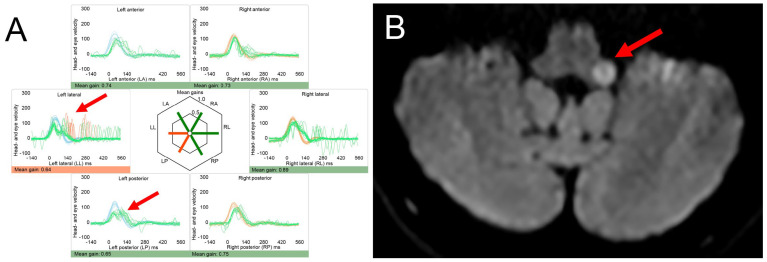
Illustrative case of a patient presenting with acute-onset persistent vertigo and inability to walk. This 51-year-old female patient with a 48 h history of acute-onset and persistent vertigo, inability to walk and vomiting presented to the ED. On examination, horizontal gaze-evoked nystagmus was observed. The head-impulse test was abnormal to the left (demonstrating a catch-up saccade as confirmed on video-head-impulse testing for the left lateral and left posterior canal [Panel (**A**), marked with red arrows]). The patient did not exhibit a skew deviation or spontaneous nystagmus. She was unable to stand or sit unassisted (consistent with grade 3 truncal instability [89]). Diffusion-weighted imaging on MRI revealed a discrete swelling lesion in the left dorsolateral medulla (Panel (**B**), marked with a red arrow). The patient had supratentorial lesions and showed positive oligoclonal bands on lumbar puncture, finally diagnosed with relapsing-remittent MS. The patient was then started on ocrelizumab. For video-head-impulse testing (in panel (**A**)) eye velocity traces (in green) and head velocity traces (in red for testing the right vestibular organ and in blue for testing the left vestibular organ) are plotted against time. Eye velocity traces were inverted for better visualization and comparison with the head velocity traces. Gain was calculated as the ratio of the area under the de-saccaded eye-velocity curve to the area under the head-velocity curve, corresponding to a de-saccaded position gain. A summary plot in the center shows average individual vestibulo-ocular reflex (VOR)-gains ± 1 standard deviation for all six semicircular canals.

**Figure 3 brainsci-15-00995-f003:**
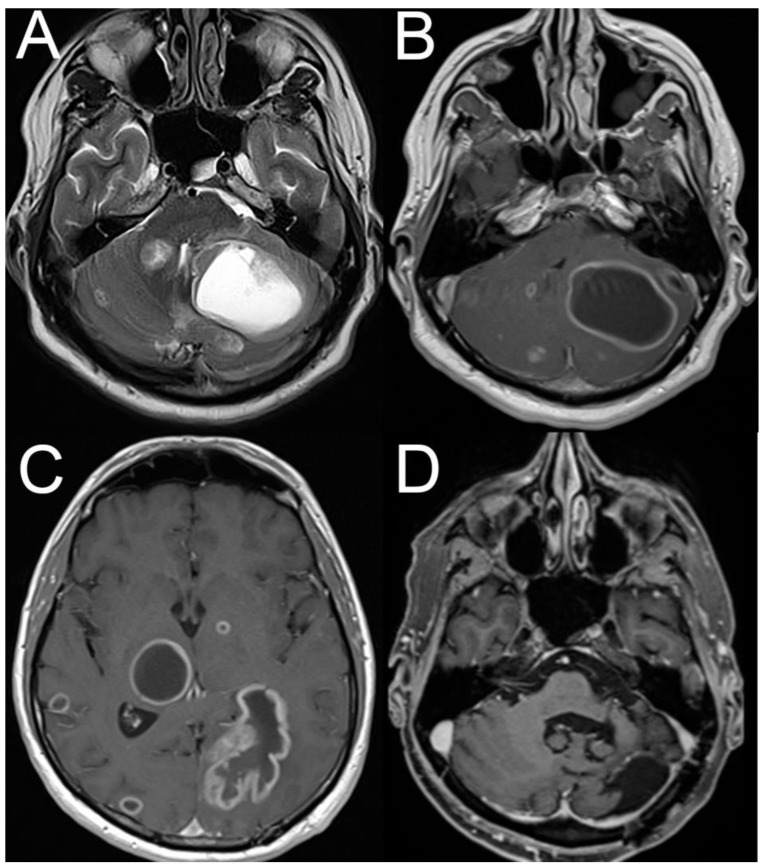
Illustrative case of a patient presenting with acute-onset positional vertigo and gait imbalance initially misdiagnosed as benign paroxysmal positional vertigo [103]. A 44-year-old male patient reported new-onset, position-dependent vertigo with nausea and gait imbalance for 10 days. He presented with a geotropic horizontal positional nystagmus and vertigo, suggestive of lateral-canal benign paroxysmal positional vertigo (BPPV). However, the repeated repositioning maneuvers did not relieve the patient’s symptoms. On magnetic resonance imaging (MRI) axial T2-weighted (panel (**A**)) and axial T1 post contrast (panel (**B**)) images demonstrate multiple cystic cerebellar space-occupying lesions compressing the fourth ventricle. Additionally, multiple supratentorial cystic mass lesions were depicted on MRI (panel (**C**), axial T1 post contrast sequence). Urgent surgical resection revealed a histopathologic diagnosis of a metastatic tumor from adeno-carcinoma of the lung. At follow-up after 9 months (panel (**D**)) a status post left-sided sub-occipital craniotomy and resection of the large left-cerebellar cystic mass lesion can be seen. This figure has been taken from a previous publication that is licensed under a Creative Commons Attribution 4.0 International License [103]. (Figure source: Department of Radiology, Cantonal Hospital of Baden, Baden, Switzerland).

## Data Availability

No new data were created or analyzed in this study. Data sharing is not applicable to this article.

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
