# Peer review of "Acute Vertigo, Dizziness and Imbalance in the Emergency Department—Beyond Stroke and Acute Unilateral Vestibulopathy—A Narrative Review"

_brainsci, 2025, doi:10.3390/brainsci15090995_

Round 1

Reviewer 1 Report

Comments and Suggestions for Authors

Dear Ladies and Gentlemen, Dear Journal-Team,
the intersting mnuscript 'Acute vertigo, dizziness and imbalance in the emergency department-beyond stroke and acute unilateral vestibulopathy - a critical review' helps in the sometimes necessary fast distinguishing between vertigo reasons. It is well written. The tables and figures are sufficient.
a) Please correct in line 128, as vertigo caused by acute vestibular hair cell damage is not persistent. In Meniere's disease the vestibular and auditory hair cell damage is progressing with auditory and vestibular episodes and vestibualar symptom free intervals due to central neuroplasticity, which usually compensates peripheral and central vestibular deficits in days to weeks, with sometimes resting deficits under vestibular stress. Therapy in disturbing cases of Meniere's disease is the ablation of the vestibular hair cells, e.g. by transtympanic gentamicin application, as gentamicin is about 4 times more vestibulotoxic than cochleotoxic.
b) The term vestibular neuritis in line 228 is often equivalently used to the term vestibular neuropathy. But as infection is less often the cause than vascular reasons and difficult to prove, the more unspecific term vestibular neuropathy is preferred.
c) As vertigo and dizziness involves many numerous specialities, Table 1 would benefit by marking the various specialitis (neurology, ENT, cardiology, internal medicine, orthopaedics, psychiatry) and compare with Table 2 of: Inner ear symptoms and disease: Pathophysiological understanding and therapeutic options, Med Sci Monit 2013; 19:1195-1210, DOI:10.12659/MSM.889815.
d) Language: Please change in line 301 to 'complete', in line 409 to 'by vestibular'. Check the punctuation in line 465 and the style of the reference numbers in line 465 and 467.
e) Please check the references for uniform small or capital letter use in the article title according to the Journal Style Guidelines. Check reference 71 by Nam et al. for hyphen usage in the article title.
f) Please explain the abbreviation ICT in the funding section.
Sincerely,

Author Response

Dear Ladies and Gentlemen, Dear Journal-Team, the intersting mnuscript 'Acute vertigo, dizziness and imbalance in the emergency department-beyond stroke and acute unilateral vestibulopathy - a critical review' helps in the sometimes necessary fast distinguishing between vertigo reasons. It is well written. The tables and figures are sufficient.

Reply by the authors: We thank the reviewer for his/her overall positive feedback.

a) Please correct in line 128, as vertigo caused by acute vestibular hair cell damage is not persistent. In Meniere's disease the vestibular and auditory hair cell damage is progressing with auditory and vestibular episodes and vestibular symptom free intervals due to central neuroplasticity, which usually compensates peripheral and central vestibular deficits in days to weeks, with sometimes resting deficits under vestibular stress. Therapy in disturbing cases of Meniere's disease is the ablation of the vestibular hair cells, e.g. by transtympanic gentamicin application, as gentamicin is about 4 times more vestibulotoxic than cochleotoxic.

Reply by the authors: We have rephrased the sentence under discussion to improve clarity and also added another citation to provide more detail on the mechanism of action of aminoglycosides: “Bilateral vestibulopathy caused by aminoglycoside-related vestibular hair cell damage {Rivetti, 2023 #116} may manifest rapidly with acute vertigo accompanied by hearing loss [21] or emerge slowly in these situations with gait imbalance being the presenting symptom [14,20,22].»

b) The term vestibular neuritis in line 228 is often equivalently used to the term vestibular neuropathy. But as infection is less often the cause than vascular reasons and difficult to prove, the more unspecific term vestibular neuropathy is preferred.

Reply by the authors: We thank the reviewer for pointing this out. Indeed, the term vestibular neuritis can be misleading as it is linked to a specific pathophysiology (inflammation). Therefore, the preferred term according to Barany Society diagnostic criteria (Strupp M, Bisdorff A, Furman J, Hornibrook J, Jahn K, Maire R, Newman-Toker D, Magnusson M. Acute unilateral vestibulopathy/vestibular neuritis: Diagnostic criteria. J Vestib Res. 2022;32(5):389-406) is acute unilateral vestibulopathy. We have adjusted the manuscript accordingly, replacing vestibular neuritis by the term acute unilateral vestibulopathy.

 c) As vertigo and dizziness involves many numerous specialities, Table 1 would benefit by marking the various specialitis (neurology, ENT, cardiology, internal medicine, orthopaedics, psychiatry) and compare with Table 2 of: Inner ear symptoms and disease: Pathophysiological understanding and therapeutic options, Med Sci Monit 2013; 19:1195-1210, DOI:10.12659/MSM.889815.

Reply by the authors: We thank the reviewer for his/her suggestion. Following the recommendation of the other reviewers, we have updated table 1 in our manuscript, now comparing neurological vs. non-neurological disorders. However, we focused on the frequency of presentation rather than on the specialty primarily involved, since there will be substantial overlap in the specialty first evaluating the patient depending on presenting symptoms. 

d) Language: Please change in line 301 to 'complete', in line 409 to 'by vestibular'. Check the punctuation in line 465 and the style of the reference numbers in line 465 and 467.

Reply by the authors: we thank the reviewer for spotting these mistakes!

e) Please check the references for uniform small or capital letter use in the article title according to the Journal Style Guidelines. Check reference 71 by Nam et al. for hyphen usage in the article title.

Reply by the authors: We thank the reviewer for bringing this up. We have updated citation 71 (Nam et al.). Regarding small / capital letter use we will resolve this with the editorial office.

f) Please explain the abbreviation ICT in the funding section.

Reply by the authors: we have added the explanation. "ICT" stands for "Information and Communication Technology".

Reviewer 2 Report

Comments and Suggestions for Authors

The manuscript entitled “Acute vertigo, dizziness and imbalance in the emergency department beyond stroke and acute unilateral vestibulopathy: a critical review” addresses a highly relevant and multidisciplinary topic. Vertigo and dizziness are among the most frequent reasons for consultation in emergency departments, and having a synthesis that explores non-vascular and non-vestibular causes is of great value for clinical practice. The paper is clearly structured, with well-differentiated sections and supported by useful tables and figures. Moreover, the integrative approach adopted by the authors represents a valuable contribution for the clinical reader.

However, before publication, I believe that several improvements are needed to further strengthen the scientific quality and practical applicability of the manuscript:

  1. Review methodology: Although this is not a systematic review, it would be advisable to include a brief methodological section specifying which databases were consulted, the general criteria for literature selection, and the approximate time frame considered. This would increase transparency and reproducibility, without the need to strictly follow PRISMA.
  2. Critical appraisal of the evidence: Much of the data included comes from small series, case reports, or previous narrative reviews. Explicitly highlighting the strength and limitations of the evidence in each section would help readers to better assess the reliability of the information presented.
  3. Updating the references: Some relatively old references (2007–2008) are used to support prevalence data and the distribution of etiologies. It would be advisable to complement these with more recent literature (from the last 5–10 years) to confirm or update these epidemiological findings.
  4. Tables and figures: The differential diagnosis table is a strong feature of the manuscript and should be maintained in its current form regarding content. To further enhance its clinical usefulness, I suggest:
    • Improving readability with clearer subheadings or a visual separation between neurological and non-neurological causes.
    • Adding a category for musculoskeletal causes, such as cervical disorders (cervicogenic dizziness) and temporomandibular disorders, which have been associated in some studies with dizziness and imbalance. Although the evidence is limited and sometimes controversial, including these conditions would provide a more comprehensive overview.
    • Clarifying in the text that while the table is intentionally comprehensive, in clinical practice clinicians should prioritize the most prevalent and potentially dangerous conditions.
  5. Clinical applicability: It would be valuable to reinforce the practical usefulness for emergency physicians by adding recommendations on how to prioritize diagnostic tests and which red flags should alert clinicians to potentially serious conditions.

In conclusion, this is a well-prepared manuscript addressing a highly relevant clinical issue with potential for significant impact on clinical practice. With the suggested improvements in methodology, updated references, and clinical applicability, as well as the inclusion of musculoskeletal causes in the differential diagnosis, the article will achieve greater scientific robustness and practical value.

Author Response

The manuscript entitled “Acute vertigo, dizziness and imbalance in the emergency department beyond stroke and acute unilateral vestibulopathy: a critical review” addresses a highly relevant and multidisciplinary topic. Vertigo and dizziness are among the most frequent reasons for consultation in emergency departments, and having a synthesis that explores non-vascular and non-vestibular causes is of great value for clinical practice. The paper is clearly structured, with well-differentiated sections and supported by useful tables and figures. Moreover, the integrative approach adopted by the authors represents a valuable contribution for the clinical reader.

However, before publication, I believe that several improvements are needed to further strengthen the scientific quality and practical applicability of the manuscript:

Review methodology: Although this is not a systematic review, it would be advisable to include a brief methodological section specifying which databases were consulted, the general criteria for literature selection, and the approximate time frame considered. This would increase transparency and reproducibility, without the need to strictly follow PRISMA.

Reply by the authors: We supplemented the discussion accordingly and provided the PRSIMA guideline (Page Lines; Supplementary file for review). Also, we added a brief methodology section:

“We searched MEDLINE through PUBMED for English-language articles based on the following strategy and looking for specific components in all articles: (1) acute / new-onset or recurrent dizziness / vertigo caused by (2) etiology other than central or peripheral vestibular origin. There was no formal rating or study-quality assessment of identified citations and we did not apply pre-specified inclusion or exclusion criteria. Only original data on human subjects were included for this review. We did not limit our search to a specific publication period (e.g. last 10 years). A manual search of the references of articles discussed here was also performed. Research abstracts from meeting proceedings or unpublished studies were omitted.»

Critical appraisal of the evidence: Much of the data included comes from small series, case reports, or previous narrative reviews. Explicitly highlighting the strength and limitations of the evidence in each section would help readers to better assess the reliability of the information presented.

Reply by the authors: We do agree that including data from small case series or even single case reports may pose a significant limitation of the strength of evidence. We now address this with more detail in the limitations section at the end of the manuscript. While adding a discussion of strength and limitation of evidence in each section would definitely provide more in-depth information about the evidence available, it is beyond the scope of this narrative review and would add substantially to the length of this manuscript, thus impacting its readability. Therefore, we would like to prefer not to add such detailed discussion to each section in favor of a general discussion at the end of the manuscript:

“Being a narrative review, this paper has several limitations that need to be considered. First, we did not perform a systematic review of the literature, therefore there is a certain risk for selection bias. However, this approach at the same time allowed us to provide a broad overview of the topic, emphasizing the spectrum of possible clinical presentations and underlying causes. Second, we did not perform a formal assessment of the quality of studies cited in this review (such as QUADAS2 or similar), thus, some findings discussed here may have low level of evidence, especially when originating from small case series or single case reports. Third, for certain underlying disorders or clinical presentations literature available is relatively old, with more recent publications lacking. Thus, diagnostic approaches described in these publications may have changed meanwhile and new diagnostic tests may have significantly altered clinical practice.»

Updating the references: Some relatively old references (2007–2008) are used to support prevalence data and the distribution of etiologies. It would be advisable to complement these with more recent literature (from the last 5–10 years) to confirm or update these epidemiological findings.

Reply by the authors: We do agree with the reviewer that up-to-date references are essential. However, for a narrow field such as neuro-otology availability of recent epidemiological data is often challenging. We have added one more recent citation (Goeldlin et al. 2019, PMID 31531764) to the introductory part, which is in line with previous work being based on large databases.

Tables and figures: The differential diagnosis table is a strong feature of the manuscript and should be maintained in its current form regarding content. To further enhance its clinical usefulness, I suggest:

    • Improving readability with clearer subheadings or a visual separation between neurological and non-neurological causes.

Reply by the authors: We have revised Table 1 accordingly, including columns for both neurological and non-neurological causes.

Adding a category for musculoskeletal causes, such as cervical disorders (cervicogenic dizziness) and temporomandibular disorders, which have been associated in some studies with dizziness and imbalance. Although the evidence is limited and sometimes controversial, including these conditions would provide a more comprehensive overview.

Reply by the authors: We have updated Table 1 accordingly, adding a category for musculoskeletal causes.

Clarifying in the text that while the table is intentionally comprehensive, in clinical practice clinicians should prioritize the most prevalent and potentially dangerous conditions.

Reply by the authors: We added a comment accordingly in the main text: “Notably, clinicians should prioritize the most prevalent and potentially dangerous conditions.»

Clinical applicability: It would be valuable to reinforce the practical usefulness for emergency physicians by adding recommendations on how to prioritize diagnostic tests and which red flags should alert clinicians to potentially serious conditions.

Reply by the authors: We thank the reviewer for his/her suggestion. We do provide a brief summary on preferred diagnostic tests and also have added a statement in the conclusions regarding red flags. However, this falls short of a diagnostic algorithm that would guide the clinician through the diagnostic workup of the patient.

“Finding the dangerous disorders is a priority, with rapidly worsening symptoms, deteriorating vital signs, impaired consciousness and emerging focal neurologic signs reflecting red flags» 

In conclusion, this is a well-prepared manuscript addressing a highly relevant clinical issue with potential for significant impact on clinical practice. With the suggested improvements in methodology, updated references, and clinical applicability, as well as the inclusion of musculoskeletal causes in the differential diagnosis, the article will achieve greater scientific robustness and practical value.

Reply by the authors: we thank the reviewer for his/her detailed feedback!

Reviewer 3 Report

Comments and Suggestions for Authors

Thank you for the opportunity to review the document entitled “Acute vertigo, dizziness and imbalance in the emergency department – beyond stroke and acute unilateral vestibulopathy a critical review”, which was aimed to “review the diagnostic approach to patients with acute onset dizziness in the emergency room and discuss the most important differential diagnoses beyond stroke and acute unilateral vestibulopathy”.

The topic is of interest for diverse disciplines and the document comprises a wide  narrative review; however, some concerns preclude its publication as presented.

The title categorizes the research as a critical review. However, the  document describes a more general narrative review. A critical review should demonstrate  extensive research and critical evaluation of its quality; to favour clinical applications, a description on both the literature search strategy and the consulted resources are essential; It should go beyond summary, to include analysis and conceptual innovation, with perspectives and limitations.

Since the topic imply a potentially wide readership, not always familiar with neurotology, an introduction to the  clinical definitions of the main symptoms and signs that are discussed In the document is required; as well as a short introductory paragraph on the main vestibular causes for seeking acute medical care.

The reader may benefit with more information about the accompanying symptoms and signs that may help on the differential diagnosis.

Please consider that:

- oncologic emergencies (not exclusive of posterior fossa tumours) can occur at any time during the course of a malignancy (including treatment) and even years after surveillance; while  first presentation of a brain tumour as an acute event is not so rare. Neuro-oncologic patients are at high risk of developing multiple complications such as intracranial hypertension, brain herniation, intracranial bleeding, and other emergencies. Situations that usually need rapid decision-making and management.

- different effects of pregabalin and gabapentin

- the carbamazepine prescription for diabetes neuropathy instead of epilepsy.

- symptoms due to depression medication withdrawal.

- more information is required on vitB12 deficiency, particularly in the elderly.

Conclusion are written like a summary, which can be described in a Table, while the reader could get a final message on the authors perspective that is  supported by the review.   

Table 1 could be revised to include uncertainty. Since some of the diagnosed described as benign could be not so benign and require prompt attention.

Figures resolution needs to be improved, particularly Figure 2.

Author Response

Thank you for the opportunity to review the document entitled “Acute vertigo, dizziness and imbalance in the emergency department – beyond stroke and acute unilateral vestibulopathy a critical review”, which was aimed to “review the diagnostic approach to patients with acute onset dizziness in the emergency room and discuss the most important differential diagnoses beyond stroke and acute unilateral vestibulopathy”.

The topic is of interest for diverse disciplines and the document comprises a wide narrative review; however, some concerns preclude its publication as presented.

The title categorizes the research as a critical review. However, the document describes a more general narrative review. A critical review should demonstrate extensive research and critical evaluation of its quality; to favour clinical applications, a description on both the literature search strategy and the consulted resources are essential; It should go beyond summary, to include analysis and conceptual innovation, with perspectives and limitations.

Reply by the authors: We thank the reviewer for pointing this out. Indeed, this review matches better the description of a narrative review than a critical review. Thus, we have changed the title accordingly.

Since the topic implies a potentially wide readership, not always familiar with neurotology, an introduction to the clinical definitions of the main symptoms and signs that are discussed in the document is required; as well as a short introductory paragraph on the main vestibular causes for seeking acute medical care.

Reply by the authors: We now provide an additional paragraph in the introduction:

«Understanding the common neurological and neurotological causes of acute dizziness/vertigo can be the framework for triaging the patients. Vestibular neuritis or acute unilateral vestibulopathy (VN/AUVP) can be the common cause. As those accounts for the majority of the population of acute dizziness, nearly 5–10% of patients of posterior circulation stroke can also present with acute dizziness [7]. Benign paroxysmal positional vertigo, Menière disease, or vestibular migraine usually present with triggered or recurrent vestibular syndrome, but it can often confuse the diagnosis in case of first presentation. Catching the nystagmus and neurotological signs indicating peripheral or central vestibulopathy is essential for the diagnosis in these cases [8-10].»

The reader may benefit with more information about the accompanying symptoms and signs that may help on the differential diagnosis.

Reply by the authors: As suggested by the reviewer, we now provide more detailed information about the clinical assessment in the introductory part of the review. This is in addition to the differential diagnoses already discussed in the specific chapters:

“While new-onset or recurrent vertigo, dizziness or gait imbalance is the leading symptom of the spectrum of disease discussed here, knowledge about accompanying signs and symptoms, conditions triggering or worsening complaints and pre-existing conditions including current medication is essential in the diagnostic workup.»

“Therefore, a general neurologic examination should be combined with the search for subtle ocular motor signs, an assessment of stance and gait, hearing and positional testing. While the presence of severe gait and truncal instability, central ocular motor signs (including gaze-evoked nystagmus, and skew deviation) or new-onset unilateral hearing loss make central (mostly ischemic) vestibular causes more likely, triggered nystagmus on positional testing favors benign paroxysmal positional testing. Likewise, absence of ocular motor, vestibular abnormalities and focal neurologic signs make non-neurological causes more likely (as discussed in the following sections).»

Please consider that:

- oncologic emergencies (not exclusive of posterior fossa tumours) can occur at any time during the course of a malignancy (including treatment) and even years after surveillance; while first presentation of a brain tumour as an acute event is not so rare. Neuro-oncologic patients are at high risk of developing multiple complications such as intracranial hypertension, brain herniation, intracranial bleeding, and other emergencies. Situations that usually need rapid decision-making and management.

Reply by the authors: We thank the reviewer for pointing out this important aspect. We revised Table 1 accordingly, adding neuro-oncological emergencies to the urgent causes in the patient presenting with acute vertigo/dizziness.

- different effects of pregabalin and gabapentin

Reply by the authors: We do agree with the reviewer that pregabalin and gabapentin differ in several aspects, including the range of side-effects reported (with somewhat less side effects in pregabalin compared to gabapentin), pharmacokinetics and treatment efficacy (with pregabalin being more favorable). However, discussing differences regarding efficacy and tolerability between these two substances is beyond the scope of this review. We would like to focus on the statement that both substances may cause vertigo or dizziness as a side-effect.

- the carbamazepine prescription for diabetes neuropathy instead of epilepsy.

Reply by the authors: We clarified on this in the revised manuscript:

These include antiepileptic drugs often used to treat epilepsy or neuropathy (especially phenytoin, carbamazepine [17], and oxcarbazepine, but also lacosamide, gabapentin, pregabalin) [18], benzodiazepines and lithium.»

- symptoms due to depression medication withdrawal.

Reply by the authors: We thank the reviewer for bringing this up. We now also point to potential anti-depressant medication withdrawal effects:

“The less abrupt onset and accompanying autonomic signs and symptoms (as in alcohol, antidepressants or opiate withdrawal) may help differentiate drug-withdrawal-related vertigo from vertebrobasilar stroke or acute unilateral vestibulopathy (AUVP) [11].”

- more information is required on vitB12 deficiency, particularly in the elderly.

Reply by the authors: We have added a brief section on Vitamin B12 deficiency, albeit in most cases with Vitamin B12 deficiency symptom onset will be gradual rather than acute. Thus, initial presentation in the ED setting is unlikely.

“Vitamin B12 deficiency may lead to vertigo/dizziness or gait imbalance by several mechanisms, including sensory ataxia due to peripheral polyneuropathy or myelopathy (being rarely acute-onset) and orthostatic hypotension due to anemia. Treatment focuses on Vitamin B12 supplementation in these cases.»

Conclusion are written like a summary, which can be described in a Table, while the reader could get a final message on the authors perspective that is supported by the review.   

Reply by the authors: The conclusions were written in the intention to provide the take-home messages in brief for the reader. Thus, the reviewer’s perception was absolutely right! Admittedly, the perspective as mentioned by the reviewer was missing (thank you very much for spotting this!). Thus, we have added an outlook section at the end of the conclusions:

“Evaluating the acutely dizzy patient in the ED-setting remains challenging. The difficulty may stem from the broad range of underlying disorders and the complexity of diagnostic algorithms available, and the technical demands of bedside tests and interpretation. Therefore, dedicated training in the management of the acutely dizzy patient for ED physicians should be prioritized. Until such time that emergency clinicians receive training in the physical examination techniques used to diagnose patients with acute dizziness, vertigo and imbalance, wider use of telemedicine services, ideally with video-oculography is recommended although this too is difficult to implement at scale.”

Table 1 could be revised to include uncertainty. Since some of the diagnosed described as benign could be not so benign and require prompt attention.

Reply by the authors: We thank the reviewer for this suggestion. Based on the feedback of the other reviewers we have revised table 1 to distinguish between neurological and non-neurological disorders and benign / dangerous conditions. Furthermore, we assigned specific diseases to different categories based on their estimated incidence. Adding information on the certainty of given diagnoses would significantly increase the complexity of the table. Thus, we tentatively suggest not to add this information. We have also re-reviewed Table 1 regarding classification and have made adjustments such as moving neuro-oncological emergencies to the “dangerous” category.

Figures resolution needs to be improved, particularly Figure 2.

Reply by the authors: For figure 2B resolution is limited by the constraints of this MRI-sequence (diffusion-weighted imaging) and the focus on the cerebellum / brainstem. Unfortunately, we cannot improve the resolution of this figure. With regards to Figures 1, 2A and 3, high-resolution (300dpi) figures have been submitted separately also. Thus, we are confident that the resolution issue will be resolved.

Round 2

Reviewer 2 Report

Comments and Suggestions for Authors

I appreciate the efforts made by the authors in this revised version of the manuscript. After reviewing the changes incorporated, I consider that all the observations raised in the first round have been satisfactorily addressed. The manuscript has improved in clarity, coherence, and scientific quality.

Therefore, I have no further comments and recommend the acceptance of the article for publication in the journal.